# Growth promotion and antibiotic induced metabolic shifts in the chicken gut microbiome

Germán Plata [1✉], Nielson T. Baxter [1], Dwi Susanti[1], Alyssa Volland-Munson[1], Dharanesh Gangaiah[1], Akshitha Nagireddy[2], Shrinivasrao P. Mane[1], Jayanth Balakuntla[2], Troy B. Hawkins[1] & Arvind Kumar Mahajan[1]

Antimicrobial growth promoters (AGP) have played a decisive role in animal agriculture for over half a century. Despite mounting concerns about antimicrobial resistance and demand for antibiotic alternatives, a thorough understanding of how these compounds drive performance is missing. Here we investigate the functional footprint of microbial communities in the cecum of chickens fed four distinct AGP. We find relatively few taxa, metabolic or antimicrobial resistance genes similarly altered across treatments, with those changes often driven by the abundances of core microbiome members. Constraints-based modeling of 25 core bacterial genera associated increased performance with fewer metabolite demands for microbial growth, pointing to altered nitrogen utilization as a potential mechanism of narasin, the AGP with the largest performance increase in our study. Untargeted metabolomics of narasin treated birds aligned with model predictions, suggesting that the core cecum microbiome might be targeted for enhanced performance via its contribution to host-microbiota metabolic crosstalk.

[1] Discovery research, Elanco Animal Health, Greenfield, IN 46140, USA. [2] Discovery research, Elanco Animal Health, Bangalore 560008, India.
✉email: german.plata_caviedes@elancoah.com

The discovery in the 1940's that feeding of antimicrobial compounds could accelerate animal growth played a significant role in shaping and expanding animal agriculture to its current scale[1,2]. Intensive animal food production has sustained the increasing global demand for animal protein[3] but has also raised concerns about selection for antimicrobial resistance (AMR) leading to antimicrobial-resistant infections in human and animal clinical settings[4–7]. Over 70% of antimicrobials produced globally are used in animal agriculture[8] and although the use of antimicrobial compounds for growth promotion is being banned in multiple countries[9], their application has been projected to increase due to increased adoption of large-scale intensive farming operations in countries such as Brazil, Russia, India, China, and South Africa[10]. The impact of antimicrobial growth promoters (AGP) has been especially important for the poultry industry, which has expanded more than any other animal protein source over the past half-century[11]. Among poultry species, chickens (Gallus gallus) account for most of the global poultry production. Thus, there is a high incentive to identify AGP alternatives that can sustain the increasing demand for poultry while reducing the risk for AMR selection and its consequences for human and animal health.

Despite their widespread use, there are multiple gaps in our understanding of how AGP impact animal performance[12–14]. This is significant as knowledge of the physiological mechanisms is likely to uncover biological targets to induce the same benefits of AGP without the risk of AMR[13]. Multiple antimicrobial compounds have been used to increase weight gain and feed efficiency in farm animals. These include cyclic peptides (e.g., bacitracin), ionopohores (e.g., monensin, narasin), streptogramins (e.g., virginiamycin), orthosomycins (e.g., avilamycin), and macrolides (e.g., tylosin, spiramcycin) among others[15,16]. Even though these compounds differ in their antimicrobial spectrum and antibacterial mode of action[17], whether they increase performance through similar or different mechanisms is not clear. Several hypotheses have been advanced to explain how AGP may contribute to host physiology[15,18], nevertheless, compound-specific mechanisms, if any, remain poorly described. Some of the generic AGP proposed mechanisms include limiting opportunistic pathogens and subclinical infections[19], decreased microbial competition for host nutrients[20], modulating host fat digestion and utilization[21–23], inhibiting the production of toxins in the gut[24], regulating host's immunity and inflammation[12,25], and improving nitrogen balance[26]. Given the involvement of the gut microbiome in most of these proposed mechanisms, engineering the gut microbiome to drive the performance and health status of animals has received considerable attention in the fight against AMR[27,28].

The avian caeca are a pair of sacs at the transition between the ileum and the large intestine. Among all organs, the cecum contains the highest density of microbial biomass in chickens[29], and it has been implicated in processes including fermentation of undigested fiber, nitrogen recycling, water absorption, and overall nutritional status[30,31]. Analysis of the composition of cecal microbial communities in broiler chickens has shown a predictable progression through development towards a stable microbial composition[32]. Additionally, several studies have identified correlations between members of the cecal microbiome and bird performance under antibiotic treatment and antibiotic-free conditions[32–34], as well as similarities in the microbiome effects induced by antibiotics and probiotics[35]. While those associations support the notion that manipulating the composition of the gut microbiome can provide a path for replacing AGP, few studies have assessed the overlap in the taxonomic and functional consequences of distinct AGP[36,37] to guide microbiome engineering efforts.

To advance our understanding of the possible roles of the cecal microbiome on AGP mechanisms, here we used four distinct AGP to investigate functional changes in the cecal microbiome of broiler chickens. First, we describe how AGP leading to different levels of growth promotion alter the community structures and gene abundance profiles of the cecal microbiomes of treated birds. Second, we demonstrate the contribution of core microbiome members to many of the observed functional shifts. Third, to investigate how these microorganisms may affect host physiology, we generate draft metabolic reconstructions for the core microbial genera of the cecal microbiome and model and compare phenotypic traits of the corresponding communities in silico. Finally, we use untargeted metabolomics analysis to complement the metagenomics and metabolic modeling results and further validate the functional impact of one of the AGP on growth performance.

## Results

### AGP significantly improved weight gain and feed conversion efficiency of broilers.

We carried out a clinical study including 500 broilers randomly assigned to either a control group or to one of four in-feed AGP treatment groups [bacitracin methylene disalicylate (BMD), avilamycin, virginiamycin, or narasin]. Compared to the control group, broilers receiving AGP in the diet showed increased daily weight gain and improved feed conversion efficiency (Fig. 1a, b, and Supplementary Data 1). These trends were present throughout the length of the study (35 days) but became visibly more marked during the finisher period (28 to 35 days). On the final week of the study, the average daily gain (ADG) was 9 to 14% higher in AGP treated animals compared to controls and the feed conversion ratio (FCR) was better in AGP treated groups by 3.8 to 6.2%. Among AGP, narasin had the highest average impact on performance, followed by virginiamycin, avilamycin, and BMD. We note that FCR values for day 7 birds are higher than expected, this is likely due to the feed being offered to birds in trays on the floor during the first week of the study, facilitating access to food but also causing spills.

### Different AGP lead to distinct cecal microbial community structures.

After confirming the AGP effects on performance, we investigated how the cecal microbiome might have changed along with the observed phenotypes. For this, the cecal contents from 15 birds per treatment (three from each pen) were used for 16 S rRNA amplicon microbiome profiling at days 7, 21, and 35 of the study (data was obtained for 223 samples). Across samples, a total 2937 amplicon sequence variants (ASV) were identified, with a mean of 137 ASV per sample. In agreement with previous reports[32], we observed a strong effect of bird age on microbiome composition (Fig. 1c), with samples from different days spanning the first two principal components of the dissimilarity matrix between samples (variance explained: ~62%). Independent of age, we also observed significant microbiome structure differences between samples from different treatment groups. An analysis of similarities (ANOSIM) based on the Bray-Curtis dissimilarities among 16 S profiles showed that, except for a couple of treatment pairs at 7 days, all pairwise comparisons between treatments resulted in significantly different microbiome compositions at every time-point (Fig. 1d). Interestingly, on days 21 and 35 there was a higher similarity (lower ANOSIM R scores) among control samples and samples from the AGP treatments producing the smallest effects on performance (BMD and avilamycin). Thus, larger performance gains due AGP application were accompanied by bigger changes in cecal microbiome composition.

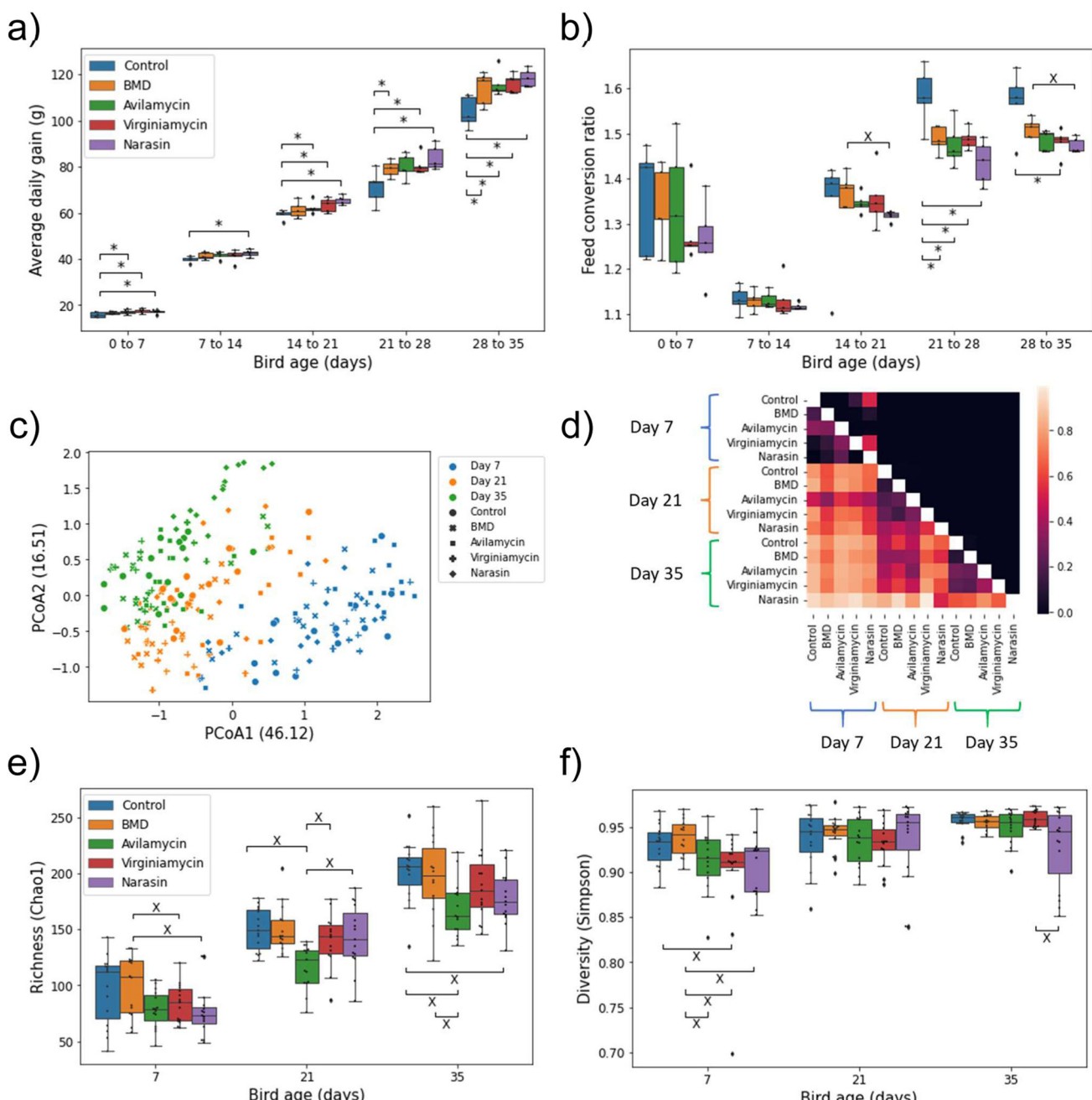

**Fig. 1 Effects of four different AGP on bird performance and cecal microbiome composition. a)** Box plot showing the average daily gain as a function of bird age. ADG was calculated separately for each week of the experiment for each of the treatment groups. **b)** Like a, but for the feed conversion ratio as a function of bird age. n = 5 pens per treatment in a and b. **c)** Principal coordinates analysis using the Bray-Curtis dissimilarity among 16 S microbiome profiles. Each symbol represents a sample. The numbers in parenthesis indicate the percentage of the dissimilarity variance explained by each of the first two principal components. **d)** ANOSIM analysis for samples from birds of different ages and AGP treatment groups. In the heatmap, values below the diagonal represent the ANOSIM R-score. Lower values indicate more similar microbial communities. Values above the diagonal indicate the corresponding p-value for the null hypothesis of similar communities. **e)** Amplicon sequence variant (ASV) richness for cecal samples from birds of different ages and AGP treatment groups quantified with the Chao1 index. **f)** Like e but for ASV diversity quantified using the Simpson index. n = 15 birds per treatment in e and f. The boxes in a, b, e and f represent the median and interquartile range; whiskers indicate the range of the distribution excluding outliers. Outliers are at least 1.5 times the interquartile range below or above the first and third quartile, respectively. *: One-sided Mann-Whitney U p-value < 0.05; X: Two-sided p-value < 0.05.

Looking at the diversity within samples (α diversity), the effect of bird age on cecal microbial community structure was reflected by an increase in both richness and evenness as birds developed (Fig. 1e, f). However, the application of AGP tended to reduce microbial richness at all three timepoints, as well as evenness (Simpson diversity) in the case of day-7 samples.

Altogether, the ANOSIM and α diversity results indicate that each AGP led to the development of a distinct cecal microbial community structure. To investigate how these mature communities may have contributed to the observed performance gains, we used shotgun metagenomics to analyze the gene content of microbes in the cecum of day 35 broilers across treatments.

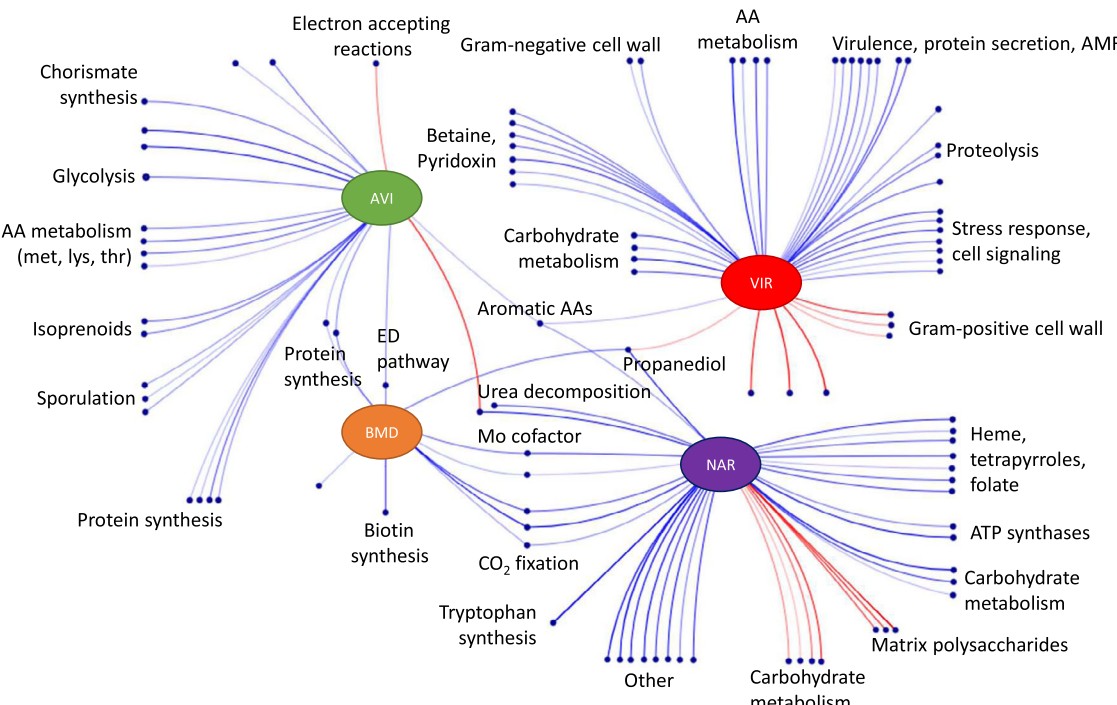

**Fig. 2 Major functional categories enriched or depleted in response to AGP relative to controls in the cecal microbiome.** Each dot represents a set of functional roles significantly enriched or depleted relative to controls (FDR < 0.15). Blue lines connect AGP to significantly enriched processes according to Gene Set Enrichment Analysis (GSEA). Red lines connect AGP to significantly depleted processes. Labels capture the common functionality in select clusters of enriched terms (see Supplementary Data 3 for the full list of terms). AVI avilamycin, VIR virginiamycin, NAR narasin, AA amino acid, ED Entner Duodoroff, Mo molybdenum.

**Partially overlapping functional pathways are enriched by different AGP.** Predicted genes from 74 day-35 samples were functionally annotated using RAST functional roles[38], and role abundances per sample were expressed in terms of mean copy numbers per genome (see Methods). Interestingly, the normalized data revealed a trend of increasing gene content per genome in the AGP treatments compared to controls, which was significant in the case of narasin (Mann-Whitney U p-value = 0.05, Supplementary Fig. 1a). Additionally, based on the Bray-Curtis dissimilarities between functional profiles we observed a trend of increasing functional distance with increasing AGP effects on performance (Supplementary Fig. 1b). Specifically, the ANOSIM R scores between profiles in the control, BMD and avilamycin groups were smaller than the corresponding distances to the virginiamycin or narasin groups. Furthermore, in agreement with the 16 S results, taxonomic profile distances calculated from the shotgun metagenomics data showed mostly significant differences between treatments (ANOSIM p-values < 0.05) and larger differences (ANOSIM R scores) for the AGP producing the highest effects on growth (Supplementary Fig. 1c). Also, there were more differentially abundant genera relative to control with virginiamycin and narasin than with avilamycin or BMD (Supplementary Data 2).

To investigate how AGP treatment altered the functional makeup of the cecal microbiome, we carried out gene set enrichment analysis (GSEA)[39] to identify subsystems or pathways enriched by each AGP compared to control birds. Figure 2 depicts the major functional processes whose abundance was enriched (blue) or depleted (red) in the cecal microbiome of AGP-treated birds. Notably, each treatment pointed to a distinct collection of enriched or depleted gene functions, with about 8-fold more functional categories changed by a single AGP than by two or more AGP treatments (Supplementary Data 3). Consistent with the ANOSIM results, there were more functional terms enriched or depleted in the virginiamycin and narasin treatments, than in birds treated with BMD or avilamycin. Among the processes independently enriched by more than one AGP were the metabolism of aromatic amino acids (all AGP except BMD), propanediol utilization, molybdenum-containing cofactor synthesis, and $CO_2$ fixation via the Wood-Ljungdahl pathway (BMD and narasin), and protein synthesis and the Entner-Doudoroff pathway (BMD and avilamycin). Processes enriched by individual AGP included biotin synthesis (BMD); methionine metabolism, glycolysis, sporulation, and isoprenoid synthesis (avilamycin); betaine and pyridoxin biosynthesis, stress response, bacterial secretion systems, antimicrobial resistance, and biosynthesis of Gram-negative cell wall components (virginiamycin); and tryptophan synthesis, urea degradation, heme and tetrapyrrole biosynthesis, and xylose utilization, among others (narasin). Narasin also led to the depletion of functional roles associated with the utilization of mucin polysaccharides as well as cell-matrix components including sialic acid, N-acetylneuraminate and hyaluronic acid (Supplementary Data 3).

In addition to gene functions enriched by their abundance difference between the AGP treatments and control, we used GSEA to look for functions enriched according to the correlation between functional role abundances and the weight at day 35 of corresponding birds within each treatment (Supplementary Data 4). In general, the gene functions enriched according to both methods were similar. Specifically, ~25% of the gene sets significantly enriched by at least one AGP compared to control, were also significantly enriched (p-value < 0.05) by a positive correlation with weight in at least one of the five treatment groups (e.g. protein synthesis, the Wood-Ljungdahl pathway, tryptophan synthesis, sporulation, glycolysis, tetrapyrrole synthesis, and urea degradation). Similarly, 50% of depleted terms with AGP compared to control showed a negative correlation with weight

in at least one treatment group. These include the utilization of sialic acid, hyaluronic acid and N-acetylneuraminate. Interestingly, while several terms were enriched by both methods for samples from the same treatment (e.g., protein synthesis in BMD or glycolysis and sporulation in avilamycin), several of the processes enriched by narasin also positively associated with weight in other treatments. Namely, tetrapyrroles and tryptophan synthesis (avilamycin), and urea degradation (control). These results suggest that these processes could potentially play a role in broiler performance even when they are induced by different treatments.

**The core microbiome accounts for a large fraction of AGP-related functional shifts**. Changes in the prevalence of genes associated with a particular function can be due to differences in either the gene content or abundance of multiple, relatively rare microorganisms or of the more prevalent and abundant members of the cecal microbiome. To check whether one of these cases dominates the functional changes observed for each AGP, we repeated the above GSEA analysis using only gene abundances calculated for core members of the microbiome. Core microbiome members were defined as the top 25 bacterial genera according to their mean relative abundance across all day-35 samples (see Methods). These core genera (Supplementary Data 5) account, on average, for ~90% of the microbial relative abundance (Supplementary Fig. 2a) estimated from the metagenomics reads and represent close to 75% of functional role relative abundances (Supplementary Fig. 2b). In the case of BMD and avilamycin, only 18 and 28% of GSEA enriched functions were still significant (p-value < 0.05) using only gene abundances from core genera. However, this number increased to 52 and 78% in the case of virginiamycin and narasin, respectively. Thus, a substantial proportion of functional changes (61 out of 117 function enrichments), especially for the most effective AGP, can be attributed to changes in the abundance or gene content of core microbiome members. We noted that the core microbiome accounts for a slightly higher fraction of relative abundances (~1-5%) in samples from the avilamycin, virginiamycin and narasin treatments compared to BMD and controls (Supplementary Fig. 2c).

Next, we asked whether functional changes across treatments were mostly driven by differences in the gene content or the abundance of the core microbial genera. For each of the 59 unique functional terms also enriched by the core microbiome we calculated the correlation across all samples between the relative abundance of core genera and their contribution to mean gene copy numbers of the corresponding functions. A positive correlation would suggest that functional differences between samples reflect abundance differences of microorganisms with a similar gene content. No correlation, on the other hand, would indicate that functional differences are due to distinct genes being carried by the same genera across samples from different treatments. As shown in Fig. 3, most correlations were found to be positive and significant, suggesting that the genus-level gene content is likely shared across treatments, and that shifts in the abundance of the corresponding functions were due to altered abundances of core genera in the cecum microbiome. We focus on the direction and not the magnitude of the correlation because although the per-sample abundance of a particular genus is estimated based on thousands of reads mapping to corresponding reference genomes, its contribution to a function is based on orders of magnitude fewer reads mapping to only a handful of genes. As an additional test, we looked at the overlap between functional roles assigned to core genera in samples from different treatments. We found (Supplementary Fig. 3) that about 80% of

the functions assigned to a particular genus in samples from one treatment were also assigned to that genus in other treatments. While this result also suggest that the genus level gene content of core genera was likely similar across all samples, we observed a few outliers. Specifically, *Salmonella*, *Escherichia*, *Clostridioides*, and *Staphylococcus* showed a much lower gene content overlap, suggesting that AGP could have enriched for different species and strains within these genera. *Salmonella* and *Escherichia* were found at much higher abundances in the virginiamycin treatment (~50% relative abundance), and they are largely responsible, along with *Staphylococcus*, for the virulence, protein secretion systems and AMR functional roles enriched by this AGP (Supplementary Fig 4 and Supplementary Note 1).

Altogether the results indicate that changes in abundance of core microbiome genera induced by AGP treatment drove a large fraction of the gene function abundance differences observed in the cecum microbiome.

**Different AMR gene abundance profiles upon treatment with distinct AGP**. To further investigate the impact of AGP on the prevalence of AMR genes in the cecum microbiome we used the $AMR++$[40] pipeline to estimate AMR gene abundances across samples. The analysis revealed a significant increase in total AMR gene abundance in the BMD, virginiamycin, and narasin treatments compared to control (Fig. 4a). Additionally, the total number of different AMR genes detected was higher in the avilamycin and virginiamycin groups (Fig. 4b). Interestingly, looking at the similarity between treatments based on their AMR gene profiles we observed similar patterns to those found when comparing taxonomic or functional profiles (Fig. 4c). Namely, the AMR gene profiles were most similar between control and AGP producing the smaller effects on performance, and least similar for virginiamycin and narasin compared to the remaining treatments. Compared to the control, the virginiamycin cecal microbiome enriched (one-sided Mann-Whitney U p-value < 0.05) for genes involved in resistance to multiple drugs (e.g., aminoglycosides, bacitracin, cationic antimicrobial peptides, elfamycins, fosfomycins, rifampin, betalactams, and aminocumarins). Narasin, on the other hand, enriched for resistance to tetracyclines. Interestingly, avilamycin reduced the abundance of genes associated with resistance to nucleosides and macrolides (Fig. 4d). In conclusion, the effect of AGP on AMR profiles was minor for BMD and avilamycin, and, similar to taxonomic and functional profiles, treatment with virginiamycin and narasin led to significantly different AMR gene abundance profiles.

**Pan-genome scale metabolic modeling of core cecum microbial genera**. We reasoned that the metabolism and growth of the most abundant members of the microbiome is likely to determine the bulk of metabolic demands and byproducts of the cecum microbiome. Specifically, metabolic fluxes driven by low abundance microbes (non-core, or <0.5% relative abundance) would have to be comparatively large to counteract the effects of core microbiome members representing 90% of microbial abundances. Given that the genome composition of these core microbes seems largely conserved across samples, we built individual metabolic reconstructions for each of the 25 core genera based on the gene functions assigned to those genera across samples (see Methods, Supplementary Data 5, and Supplementary Note 2). We then used constraints-based modeling[41] to investigate possible functional impacts of their abundance changes.

To characterize the potential impact of the metabolism of core microbes on metabolite pools in the cecum we first studied their ability to use different carbon and nitrogen sources. We simulated the ability of each model to utilize one of 129 and 76 compounds

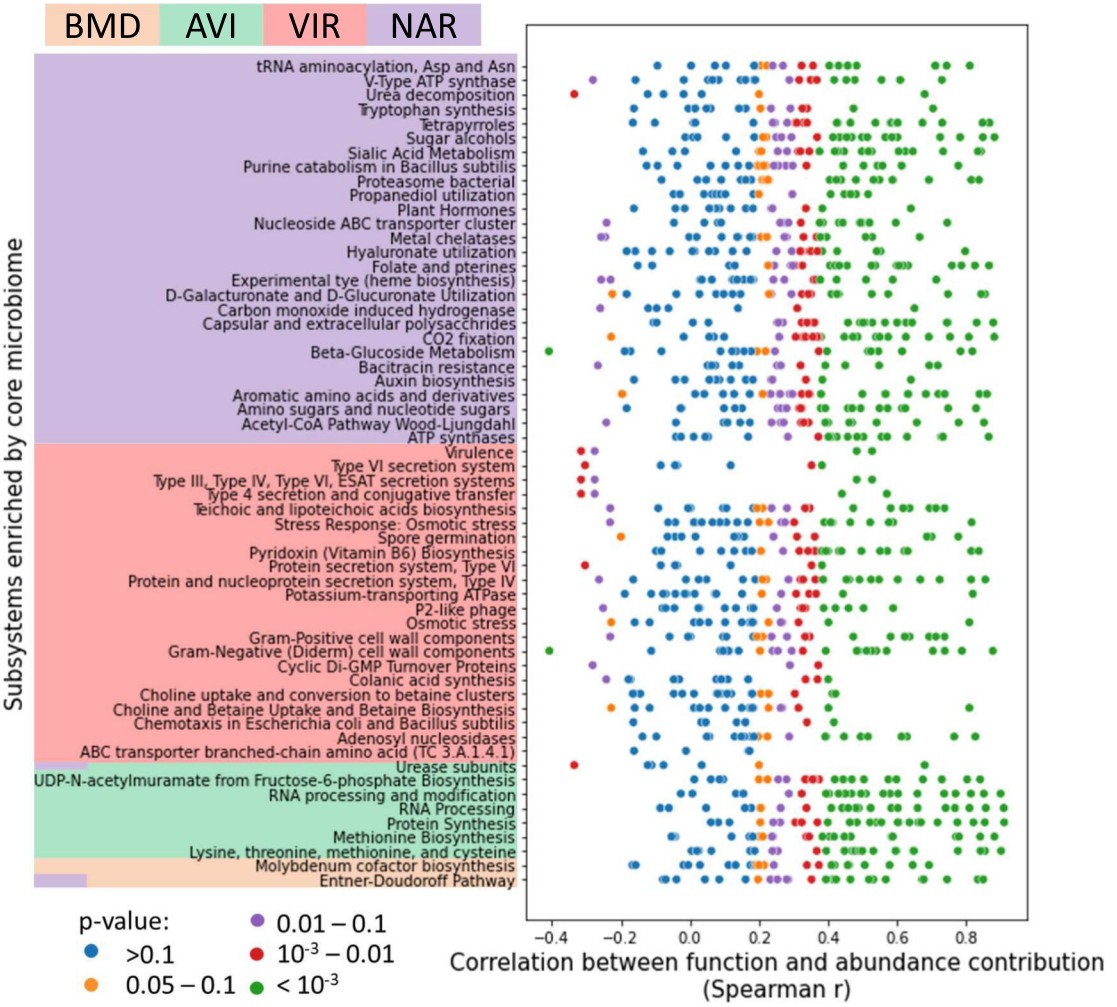

**Fig. 3 Correlation between gene and relative abundance contributions of core genera.** Rows in the figure indicate functional terms (subsystems) significantly enriched by different treatments when considering only genes assigned to core genera of the cecum microbiome. Each dot represents a core microbial genus and its corresponding value on the x-axis shows the spearman correlation between the relative abundance of the genus and its contribution to the abundance of the corresponding functional term across all samples ($n = 74$ birds). The functional contribution is calculated as the mean copy number contributed by the genus across all the genes in the subsystem. Dot colors indicate the Spearman correlation p-value. BMD bacitracin methylene disalicylate, AVI avilamycin, VIR virginiamycin, NAR narasin.

as the main carbon or nitrogen source for growth, respectively (see Methods). As shown in Fig. 5a, b, the predictions varied substantially across genera, suggesting that different genera in the cecum have unique metabolic capabilities. As expected from their larger gene sets, the reconstructions for *Escherichia*, *Salmonella*, *Clostridium*, and *Ruminococcus*, predicted a higher number of carbon and nitrogen compounds that could support growth. Given the observed differences in metabolic versatility between genera, we asked how differences in the relative abundances of these microbes across treatments may impact the metabolic potential of the core cecum microbiome. For each sample, the utilization potential of a particular carbon or nitrogen source was calculated as the relative abundance in the core microbiome of the genera able to use it for biomass synthesis. Core relative abundances were calculated by dividing the abundance of each core genus by the total abundance of core genera in each sample. In general, we found more compounds with significantly higher carbon and nitrogen utilization potentials in AGP treated birds compared to the control than compounds with lower utilization potentials (Fig. 5c, d). These results support the idea that metabolic versatility of the core cecum microbiome was increased following virginiamycin and narasin treatment.

To further explore the metabolic consequences of core microbiome composition and functional potential, we predicted essential nutrients for growth for each of the core genera (see Methods). Each genus was characterized by a mean metabolite essentiality value representing the probability that a metabolite is essential for growth across random nutritional environments. For each sample, we then calculated a total metabolic demand of the core microbiome by summing the mean metabolite essentiality of core genera weighted by their relative abundance (Fig. 5e). The results showed a significantly lower overall metabolic demand of the core microbiome in the virginiamycin and narasin treatments; that is, a lower likelihood that any single nutrient is required for the growth of a random member of the core microbiome. These results were robust to the parameters used to simulate random environments (Supplementary Fig. 8a, b) and agree with the above prediction of a higher potential to use distinct nutrients as carbon or nitrogen sources.

Interestingly, we found that the utilization potential of urea as a nitrogen source for biomass synthesis was higher in the narasin treatment (Supplementary Fig. 5a). Similarly, utilization of ammonia was predicted to be higher for both virginiamycin and narasin (Supplementary Fig. 5b). Not only was there a higher

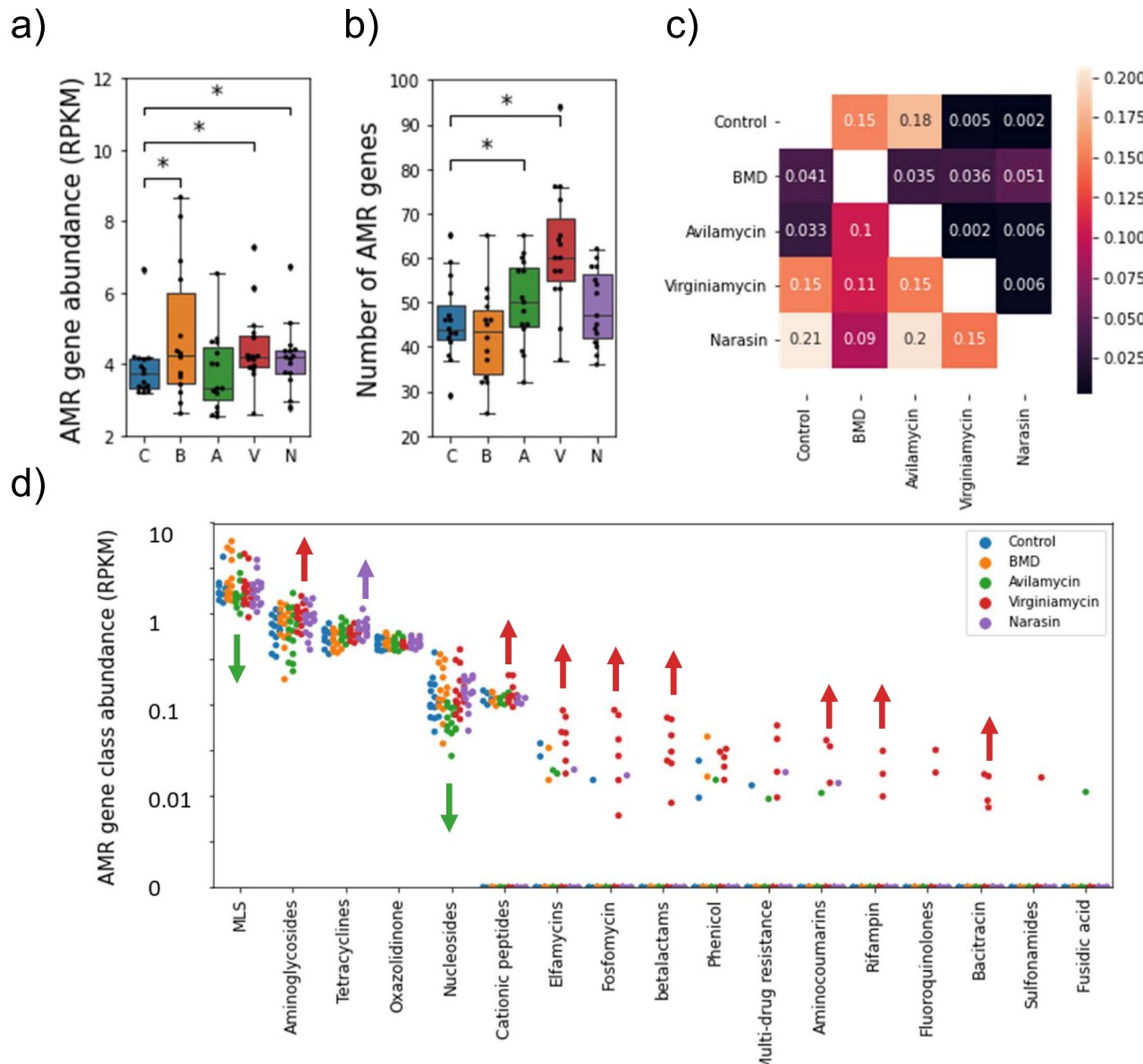

**Fig. 4 Antimicrobial resistance gene abundance in the cecal microbiome of birds treated with different AGP. a**) Total AMR gene abundance (RPKM) as a function of treatment group. **b**) Number of distinct AMR genes detected as a function of treatment group. * one-sided Mann-Whitney p-value < 0.05. $n = 15$ birds per group. The boxes in a, and b represent the median and interquartile range; whiskers indicate the range of the distribution excluding outliers. C control, B BMD, A avilamycin, V virginiamycin, N narasin. **c**) ANOSIM analysis of the Bray-Curtis dissimilarities between samples in different treatments based on their AMR profiles. In the heatmap, values below the diagonal represent the ANOSIM R-score. Lower values indicate more similar AMR profiles between treatments. Values above the diagonal indicate the corresponding p-values for the null hypothesis that AMR profiles are similar between treatments. **d**) Abundance of AMR genes of distinct classes for samples in different treatment groups (colors). Classes are sorted according to their mean AMR gene abundance across samples. Color arrows indicate a significant (p-value < 0.05) increase (pointing up) or decrease (pointing down) in the abundance of AMRs in the corresponding color treatments compared to control samples. MLS macrolides, lincosamides, and streptrogramin A and B.

predicted capacity of the core cecum microbiome to use urea and ammonia with narasin, but these metabolites were also more often essential for growth in this treatment (Supplementary Fig. 5c, d). Moreover, the genus-level metabolic reconstructions allowed us to look at the contributions of each of the core genera to the above sample-level predictions. Our analyses suggest that the higher urea utilization potential in the narasin treatment is partly driven by higher abundances of *Ruminoccocus*, *Blautia* and *Lachnoclostridium* (Supplementary Fig 5e).

**Untargeted metabolomics supports a role of the narasin-induced microbiome on nitrogen balance.** To obtain further

evidence of the metabolic consequences of cecal microbiome differences due AGP treatment, we performed untargeted metabolomics on the cecum and serum of day 35 control and narasin-treated birds. Altogether, 712 and 723 named metabolites were identified across 30 cecal and 20 serum samples, respectively. Of these, 49 were found at higher abundance and 78 at lower abundance in the cecum of narasin treated birds compared to controls (Welch's t-test, Benjamini-Hochberg corrected p-value < 0.05). In the serum, 4 metabolites were found at higher abundance and 93 at lower abundance in narasin-treated birds (Supplementary Data 6). To broadly characterize these metabolites, we ran GSEA ranking all detected metabolites according to their

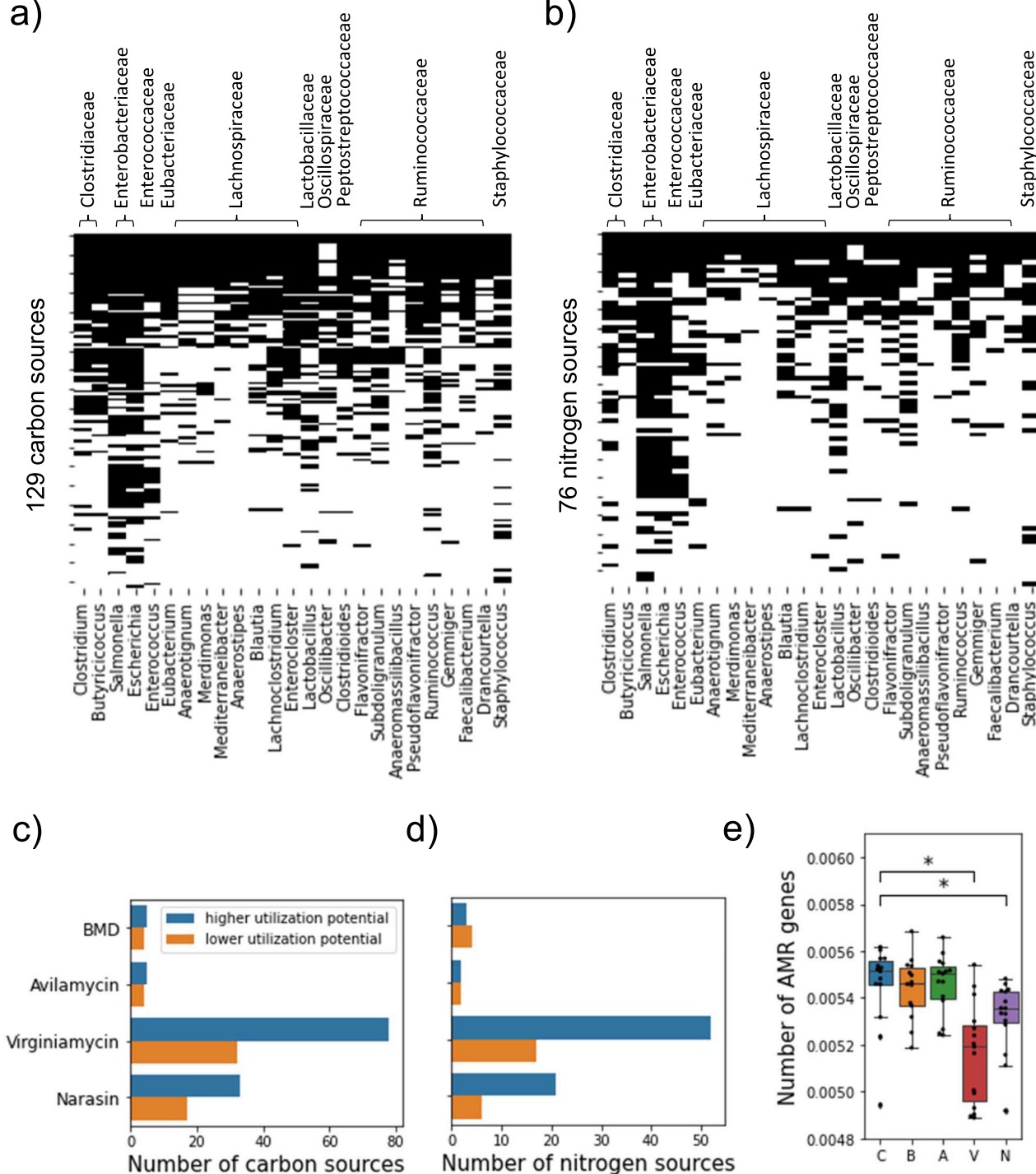

**Fig. 5 Predicted metabolic properties of the core cecum microbiome. a)** The predicted ability of each core genera to use individual compounds as main carbon sources for biomass synthesis. Core genera are shown in the columns grouped by taxonomic family. Each row represents a distinct carbon-containing compound. **b)** Like a but for the ability to use individual compounds as main nitrogen sources. **c)** The number of carbon sources with significantly (one-sided Mann-Whitney U p-value < 0.05) higher or lower utilization potential in samples from each of the AGP treatments compared to control. **d)** Like c but for compounds used as nitrogen sources. **e)** The total metabolic demand for core microbiome biomass synthesis across treatments. The total metabolic demand represents the likelihood that any given metabolite is essential for the growth of a random member of the core cecum microbiome. Boxes represent the median and interquartile range; whiskers indicate the range of the distribution excluding outliers. * two-sided Mann-Whitney U p-value < $5 \times 10^{-3}$. $n = 15$ birds per treatment. C control, B BMD, A avilamycin, V virginiamycin, N narasin.

standardized difference between treatments (see Methods). In the cecum, dipeptides (Supplementary Fig. 6a) and metabolites implicated in amino acid metabolism were significantly depleted by the narasin treatment, along with several long-chain fatty acids

(FDR < 0.15). Metabolites related to hemoglobin and porphyrin metabolism, fructose and galactose metabolism, tocopherol, inositol, riboflavin and nicotinate and nicotinamide metabolism were enriched. In the serum, phospholipids, fatty acids, and

metabolites associated with secondary bile acid metabolism, heme degradation and vitamin A metabolism were enriched, while hexosylceramides, and metabolites involved in arginine metabolism were depleted (Supplementary Data 7).

Notably, all 20 amino acids were found at lower concentrations in the cecum of narasin-treated birds (Fig. 6a). Uric acid, the main nitrogen excretion product in birds, was found at lower levels in the cecum and serum of narasin treated birds compared to controls (Fig. 6b), while urea was significantly more abundant in the serum of control birds (Fig. 6c). Thus, the abundance of several metabolites associated with altered demands or enriched functions according to the metabolic models and GSEA analyses were also found to have differential abundances in the metabolomics data. Other examples include a higher concentration in the cecum of heme degradation metabolites such as bilirubin, biliverdin and others, and the enrichment of heme and tetrapyrrole biosynthesis-related genes (Supplementary Fig. 6b), the lower demand and higher concentration of riboflavin in the cecum of narasin treated birds (Supplementary Fig. 6c), and the higher demand and lower concentration in the cecum and serum of spermidine (Supplementary Fig. 6d).

Finally, we investigated the correlation between the cecum and serum metabolite abundances and bird weights at day 35 in each of the two treatments. GSEA of metabolites ranked by the strength of their correlation with weight showed that in both treatments (control and narasin) and body sites (cecum and blood), the abundance of long-chain fatty acids tended to be negatively correlated with weight gain (Supplementary Data 8). Additionally, in the serum of narasin-treated birds and the cecum of control birds, metabolites from primary and secondary bile acid metabolism were positively correlated with performance. These observations, along with the lower concentration of long-chain fatty acids in the cecum of narasin-treated birds, suggest that lipid metabolism and absorption play a role in AGP mechanisms as previously proposed[22]. Indeed, we observed a trend of lower abundances of bile salt hydrolases in the AGP treatments compared to controls (Supplementary Fig. 7). Interestingly, amino acids and dipeptides showed contrasting enrichment patterns between the control and narasin treatments. While both types of molecules were less abundant in the cecum of narasin-treated birds, their abundance in the cecum tended to be positively correlated with growth when narasin was provided, but was negatively correlated in controls (Fig. 6d). In addition, dipeptides were positively correlated with growth when measured in the serum of control birds. A similar pattern was observed for amino acids in serum for both treatments (Fig. 6e). Thus, our observations and simulation results indicate that nitrogen metabolism in the cecum was likely altered by narasin treatment and suggest that this effect might have contributed to the nitrogen balance of the host.

## Discussion

The gut microbiome is often viewed as a virtual organ with its own development, heritability, and metabolic, immunologic, and endocrine functions[42,43]. Like other organs, the microbiome plays key physiological roles and underlies different pathologies. As AMR concerns escalate due in part to the widespread application of AGP in animal agriculture, understanding how the gut microbiome contributes to animal performance can point to novel strategies to maintain animal protein production while reducing the use of antibiotics.

Our study showed that four distinct AGP changed the composition, gene content, and AMR profile of the cecum microbiome of chickens. While several gene functions were enriched or depleted by single or multiple of the AGP treatments, additional

evidence is necessary to advance from associations such as these to hypotheses about causation. Interestingly, many of the identified shifts in the abundance of gene functions were driven by abundance changes of a reduced set of core microbiome members. This led us to hypothesize that the metabolic requirements and outputs associated with the growth of those microorganisms could contribute to the observed effects on performance. Using draft metabolic reconstructions, we estimated that the microbiome associated with AGP producing the largest effects on performance had fewer metabolite requirements for growth and were able to synthesize biomass from more diverse carbon and nitrogen sources. Thus, a high microbial density ($\sim 10^{10}$–$10^{11}$ cfu/g)[29] with a defined composition in the chicken cecum can provide a direction of causality based on the impact of microbial growth and metabolism on host physiology.

One of the functions of the avian cecum is the recycling of excreted nitrogen from the host[31]. Earlier studies showed that uric acid and urea are rapidly degraded to ammonia in the cecum[30,44]. It was also shown that nitrogen from urea supplied directly to the cecum is re-absorbed not as ammonia but as protein, urea, and amino acids[45]. Our observations of enriched urea degradation genes and lower uric acid levels in the cecum, lower urea levels in the serum, and a higher predicted core microbiome capacity to use urea and ammonia as nitrogen sources in narasin treated birds support the hypothesis that nitrogen recycling contributes to AGP effects on performance. This is also consistent with the positive correlation between urea degradation genes and bird weight at day 35 observed in control birds. A higher turnover of excreted nitrogen to usable amino acids in the cecum would support protein synthesis by the host while also reducing any potential toxicity of urea in the blood[46] (Supplementary Note 3). Moreover, because birds do not have a complete urea cycle, the higher concentration of urea in control birds could, among others, indicate a higher activity of renal arginase which could increase arginine requirements compared to birds in the narasin treatment[47,48].

In addition to nitrogen metabolism, our results showed that the concentration of primary and secondary bile salts, as well as long-chain fatty acids were altered as a function of bird weight and upon narasin treatment. This is consistent with the previously proposed hypothesis that AGP may drive performance by maintaining higher levels of active bile salts (inhibiting microbes carrying bile salt hydrolases) and thus facilitating the absorption of fatty acids[22,23]. Other processes of interest include the enrichment by narasin of heme and tetrapyrrole metabolism genes and the observation of higher levels of bilirubin and biliverdin in the cecum. Bilirubin has antioxidant and anti-inflammatory activity[49] which might have contributed to the higher performance of narasin-treated birds (Supplementary Note 4). The lower prevalence of microbial genes associated with the degradation of matrix polysaccharides observed with narasin may also point to decreased inflammation. On one hand, this could reflect reduced mucus production by the intestinal epithelium which is a symptom of low inflammation, a proposed mechanism of AGP[12]. On the flip side, lower mucus degradation by the gut microbiome could itself alleviate the inflammatory response, as increased degradation of host glycoproteins by the gut microbiota has been associated with erosion of the mucosal barrier[50].

Given that the cecal microbiome of chickens is known to differ between geographical locations[51] diets, litter quality[52] and even growth cycles[32], it is possible that the results presented here do not exactly replicate in other studies. In particular, reused litter, which is common in broiler production in the United States and other countries[53], may limit the replicability of some of our results. Nevertheless, by focusing on functional and metabolic

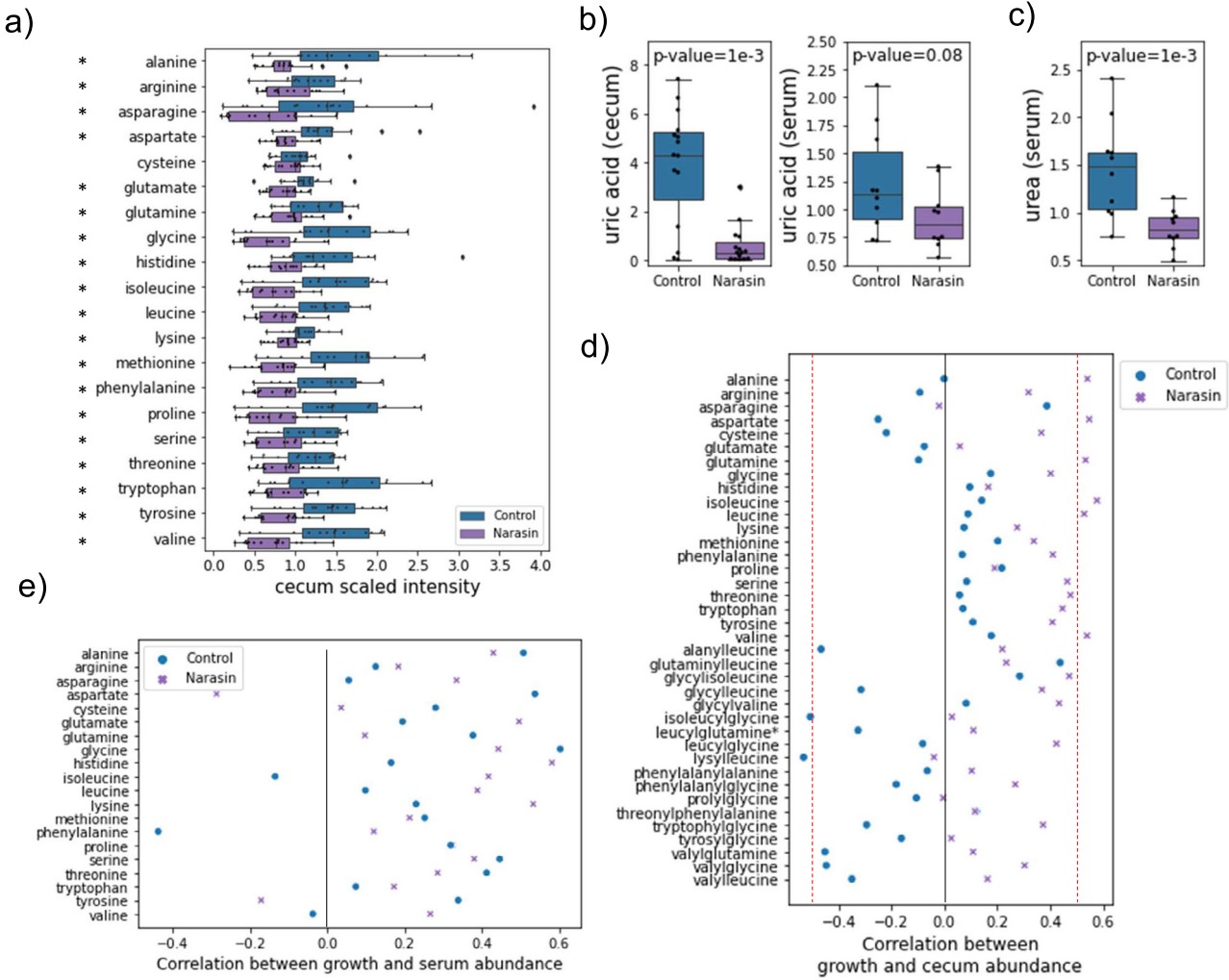

**Fig. 6 Cecum and serum metabolites altered by narasin treatment. a**) The median-normalized ion counts (scaled intensity) of 20 amino acids in the cecum of control and narasin treated birds. *Welch's t p-value < 0.05, n = 15 birds per treatment. **b**) The scaled intensity of uric acid in the cecum and serum of control and narasin treated birds. **c**) The scaled intensity of urea in the serum control and narasin treated birds. The boxes in a, b, and c represent the median and interquartile range; whiskers indicate the range of the distribution excluding outliers. **d**) The correlation between the abundance of dipeptides and amino acids in the cecum and bird weight at day 35 for control and narasin treated birds. Dashed red lines represent the correlation threshold for p-values < 0.05. **e**) The correlation between the abundance of amino acids in the serum and bird weight at day 35 for control and narasin treated birds. In panels d and e, the x-axis values show the Pearson correlation coefficient between the logarithm of the scaled intensities and weight.

changes, some of the processes described here may play a role in driving poultry performance in other settings. Indeed, a lower concentration of amino acids in the cecum was also recently observed in turkeys fed with BMD[13], and the observation of altered levels of bile salt hydrolases in our study agree with prior research on the mechanisms of AGP[22,54]. Finally, the general observation of different AGP having different impacts on the gut microbiome composition confirms prior results[36,37], while adding significantly more details as to what these changes represent in terms of functional potential and physiological impacts for the host.

In conclusion, growth promotion induced by different AGP is accompanied by common and unique cecal microbiome differences at the taxonomic, metagenomic, and metabolomic levels compared to untreated birds. Furthermore, several of these differences support mechanistic hypotheses as to how microbial activities may contribute to bird health and performance. Therefore, charting the landscape of microbiome functional changes induced by antibiotics and other types of nutritional interventions, together with a detailed description of how the

physiology of individual microbes contributes to such processes (e.g. through genome-scale metabolic modeling), sets the stage for precision microbiome engineering for both animal and human health.

## Methods

**Clinical study.** The live animal experiment and procedures were approved by Elanco institutional animal care committee, approval number IACUC # EIAC-0773.

500 healthy Ross 708 male broilers sourced from a commercial hatchery were used in the experiment. Birds had an approximate weight of 35 to 45 g at the start of the study and received a 1x dose of Coccivac®-B52 vaccine (Merck Animal Health) within 24 h of arrival at the study site. Chicks were sexed at the hatchery by feather sexing. Birds were individually tagged with a unique identifier and were randomly assigned to one of 25-floor pens (20 animals per pen). Groups of 5 pens were assigned to each of 5 different treatments at random. The 5 treatments included: 1) control birds fed least-cost diets (Supplementary Table 1) as follows: A starter diet was fed from day 0 to day 14 of the study, followed by a grower diet (14-28 d) and a finisher diet (28-35 d). 2) Birds fed the above diets supplemented with 50 g/ton BMD. 3) 20 g/ton avilamycin. 4) 16.5 g/ton virginiamycin. And 5) 70 g/ton narasin. Manufacturer and dose information is presented in Supplementary Table 2. The selected doses reflect amounts typically used for growth promotion[37]. Birds were raised for a total of 35 days in a barn on top of used litter. Litter was

obtained from untreated (control) birds from a previous study in an adjacent barn. Litter was mixed 50/50 with fresh litter and distributed to the pens for this study.

Individual body weights were collected on days 0, 7, 21, and 35. Pen weights were collected on days 14 and 28. Feed weigh backs for each pen were done on days 7, 14, 21, 28, and 35 of the study. Average bird weights and feed intake per pen were used to calculate weekly values of average daily gain and feed conversion ratios.

On days 7, 21, and 35 three birds per pen were randomly selected and euthanized for the collection of cecal contents and blood. Blood was collected via wing vein venipuncture, except for birds that were too small, in which cases direct heart puncture was used. In both cases, blood was collected into vacutainer tubes, allowed to clot, and centrifuged to obtain serum. Cecal samples were collected by aseptically squeezing the cecum content into a sterile tube. Cecal samples were placed on dry ice immediately after collection. All samples were stored at −80 °C until processing.

**16S sequencing and analysis.** Total DNA from cecal content samples was extracted using the Lysis and Purity kit (Shoreline Biome, Farmington, CT) following the manufacturer's protocol. The resulting DNA was used for library preparation with the Shoreline Biome V4 16 S DNA Purification and Library Prep Kit (Shoreline Biome, Farmington, CT). PCR amplification of the V4 region of the 16 S rRNA gene was performed using the 515 F (5′GTGGCCAGCMGCCGCGGTAA) and 806 R (5′-GGACTACHVHHHTWTCTAAT) primers. The resulting amplicons were then sequenced using $2 \times 250$ bp paired-end kits on the Illumina MiSeq platform.

Paired raw reads were processed with cutadapt (v. 2.5)[55] followed by DADA2 (v.1.12.1)[56] to generate a matrix of read counts per sample at the level of amplicon sequence variants (ASV). DADA2 parameters *maxN = 0, truncQ = 2, rm.phix = TRUE* and *maxEE = 2* were used. The *assignTaxonomy* method of DADA2 was used to assign genus-level taxonomic labels to each of the ASV based on the Silva v. 138 database[57]. Reads assigned to the Vibrionaceae family were removed from the analysis; these correspond to a small amount (~1% relative abundance) of *Vibrio harveyii* that was spiked into some of the samples. The mean sequencing depth after filtering was 43,955 reads per sample (standard deviation = 12,391).

The Chao1 and Simpson measures of alpha diversity were calculated from the matrix of raw counts using the scikit-bio (v. 0.5.6) python package and compared across treatments at each sampling time using the Mann-Whitney U test. The raw counts matrix was sum-normalized to generate a matrix of relative abundances of ASV per sample. Bray-Curtis dissimilarities between all pairs of samples were calculated from the relative abundance matrix using the *pdist* method in the sicpy (v. 1.3.1) python package. Principal component analysis of the dissimilarity matrix (principal coordinates analysis) was carried out using the implementation in the scikit-learn (v. 0.21.3) python package. ANOSIM analyses between all treatment and time point combinations were carried out using the *anosim* method in scikit-bio.

**Shotgun metagenomics sequencing and data pre-processing.** 74 frozen cecal samples from day 35 birds were sent to CosmosID (Rockville, MD) for shotgun metagenomics sequencing. DNA was extracted using the ZymoBIOMICS DNA Miniprep Kit (ZymoBIOMICS, Irvine, CA) according to the manufacturers protocol. The isolated DNA was quantified by Qbit (ThermoFisher, Coon Rapids, MN). DNA libraries were prepared using the Illumina Nextera XT library preparation kit (Illumina, San Diego, CA), with a modified protocol. Libraries were then sequenced on an Illumina HiSeq platform 2x150bp, with an average read depth of 2 million paired reads per sample.

Reads were trimmed and Illumina adapters removed using Trimmomatic (v 0.39)[58] with options *SLIDINGWINDOW:5:20 LEADING:3 TRAILING:3*. Host reads (*Gallus gallus*, GenBank assembly GCA_000002315) and spike-in reads from *Vibrio harveyii* (ATCC 14126) were removed using Bowtie2 (v. 2.3.5.1)[59].

**Taxonomic profiling from shotgun metagenomics.** Taxonomic profiling of individual reads was carried out using Kaiju (v. 1.7.3)[60] with the *nr_euk* database. Kaiju outputs for each read were expressed as the number of bases aligned to the reference database. In cases of individual reads mapping to more than one taxonomic group in the database, the number of bases assigned to each taxon was calculated as the alignment length (in base pairs) multiplied by the fraction of bases assigned unambiguously to said taxon out of all bases assigned unambiguously to all taxa to which the read mapped. The numbers of classified bases per read were used to estimate species-level relative abundances using metametamerge (v. 1.1)[61] while correcting for the average coding length per species in the *nr_euk* database. For taxonomy level analyses, only species/genera with a mean relative abundance across samples higher than $10^{-4}$ were used[62].

Differential abundance of individual genera in each of the AGP treatments compared to the control was carried out for samples collected on day 35 starting from the genus-level relative abundances. Statistical tests were done using the *taxa.compare* method in the metamicrobiomeR[63] (v. 1.2) which uses generalized additive models for location, scale, and shape with a zero-inflated beta family

distribution for the comparison of microbial relative abundances. p-values were corrected using the FDR method.

**Functional profiling.** Functional profiling of was done by first assembling reads from each sample into contigs using Spades (v. 3.13.1, option—*meta*)[64] with default parameters, followed by protein prediction with Prokka (v. 1.14.5)[65]. Predicted protein sequences were aligned to a reference database and using Diamond (v. 0.9.30)[66] and annotated based on the function assigned to the top hit in the database. To avoid misannotations[67], we only considered hits with at least 70% identity and 70% query coverage. The reference database was built starting from 46,181 complete bacterial genomes downloaded from the PATRIC[68] database representing at most 10 genomes per species from ~400 taxonomic families including those most frequently observed in broiler microbiomes. To speed up functional annotations, protein sequences within each family in the database were clustered at 90% sequence identity using CD-HIT (v. 4.8.1)[69] and the longest sequence from each cluster was kept for the diamond search.

To quantify the abundance of each function per sample, reads were mapped to the corresponding contigs using BWA (v. 0.7.17)[70], and reads mapping to each predicted protein were counted with HTseq (v. 0.11.3)[71]. Read counts per RAST functional role[38] were TPM normalized based on the Prokka protein lengths[72] and expressed in units of mean copy numbers per genome by calculating a normalization factor such that the median abundance of a set of universal-single copy bacterial genes[73] (Supplementary Table 3) was set to 1[74]. A similar procedure was used to produce a copy-number normalized matrix of functional roles across samples stratified by the taxonomic genus of the protein hits used to transfer functional annotations. This matrix was used to study the contribution of core cecum microorganisms to functional differences between samples.

ANOSIM analyses on the metagenomics derived taxonomic and functional profiles were calculated as described for the 16 S data; except that the copy number-normalized abundances were used for functional profiles. Gene set enrichment analysis (GSEA) was carried out using the *prerank* method in the gseapy (v. 0.10.2) python package. For this, gene sets were defined based on the RAST hierarchy of functional roles parsed from PATRIC genomes. For each AGP, functional roles with mean copy numbers higher than zero in at least two samples were ranked based on the Z-score of their normalized abundances compared to controls (Eq. 1)[75]:

$$Z = \frac{\mu_{AGP} - \mu_{control}}{\sqrt{\frac{\sigma^2_{AGP}}{n_{AGP}} + \frac{\sigma^2_{control}}{n_{control}}}} \tag{1}$$

where μ represent the average copy number of the roles in the respective treatment and σ represent the corresponding standard deviations. An equivalent analysis was performed by ranking functional roles with mean copy numbers higher than zero in at least three samples according to the Spearman correlation coefficient of their abundance and corresponding bird weights at day 35. The analysis was done separately for each treatment.

The core cecum microbiome was defined based on the metagenomics-based genus-level taxonomic profiles. Genera were ranked based on their mean relative abundances across samples, and the most abundant genera adding up to 90% of mean relative abundances were defined as core. The abundances of Prokka predicted genes with top diamond hits belonging to these genera were used to calculate the core microbiome contribution to functional abundances. Specifically, as the sum of the abundances of the genes with top hits in core genera. GSEA was carried out as described above for the comparison between AGP and controls but using only functional role abundances contributed by core genera.

**AMR gene abundance analysis.** AMR gene abundances per sample were determined using the *AMR++* pipeline[40] (v. 2.0.2) starting from the raw metagenomics sequences and using the MegaRes (v. 2.0.0) database. Only genes involved in drug resistance were considered. The AMR gene counts per sample obtained from *AMR++* were expressed in units of reads per kilobase per million reads (RPKM) based on the corresponding gene lengths from the MegaRes database.

**GSMM reconstruction of core genera in the cecum.** Starting from the diamond search results, we collected the RAST gene functional roles of top hits belonging to the 25 core genera. For the metabolic reconstructions we focused only on functional roles mapping to reactions in the ModelSeed database[76]. The mean number of unique metabolic roles per genus based on the metagenomics data was 398 whereas the mean number of roles across fully sequenced PATRIC genomes of the same genera was 453. Thus, we relied on the metabolic roles from reference sequenced genomes to produce lists of roles for each core genus of the expected size. Specifically, we used SiGMoiD[77], a recently developed statistical framework for modeling binary data, to estimate the conditional probability that a metabolic role not present in the annotation of a genus is present in said genus given the metabolic roles present in the annotation. We then added to each genus the topmost probable functional roles up to the mean number of metabolic roles from corresponding sequenced representatives.

Once we obtained a list of metabolic roles for each core genera, we relied on the mapping files from the ModelSeed database to generate draft metabolic reconstructions. Specifically, roles were mapped to complexes and complexes to

ModelSeed reactions to produce a list of metabolic reactions for a given genus. The template biomass compositions for Gram-positive and Gram-negative bacteria from ModelSeed were used as the biomass synthesis reactions of corresponding core genera.

Following a similar approach to that of ModelSeed[76,78], gap-filling reactions were added according to the set of reaction penalties specified in the ModelSeed database, reaction bonus scores were also calculated for each model based on the presence of reactions in RAST subsystems and "scenarios" in each of the reconstructions, as previously described[76]. A linear optimization problem was formulated starting from a universal metabolic network made from all reactions in the ModelSeed database. In the optimization, we enforced a minimal level of biomass production (0.001 mmol/g dry weight (DW)) and minimized fluxes through reactions not present in the initial reconstruction of a specific genus, weighted by corresponding reaction scores (penalties + bonuses)[79]. The optimization was done on a simulated complete media; meaning that uptake of all metabolites for which transport reactions were present in the initial reconstruction was allowed. Any additional reactions carrying non-zero flux were added to the initial reconstruction.

**Flux Balance Analysis of core genera metabolism**. We collected lists of carbon and nitrogen-containing compounds with transport reactions across all 25 reconstructions. We then used FBA to predict the ability of each of these compounds to serve as the main carbon or nitrogen source for growth as described previously[80]. Briefly, to test if a carbon source can support growth, we used FBA to predict the maximum biomass synthesis rate when the total uptake fluxes through all carbon-containing compounds were limited to 10 mM of carbon per gram of dry weight. Then for each tested compound, we simulated maximum biomass production when that compound was not included in the above constraint. If the value obtained was at least twice the baseline we considered that such compound could be used as a main carbon source. An analogous analysis was done for nitrogen sources. The utilization potential of individual compounds was calculated as the sum of the core relative abundances of genera predicted to use that compound as a carbon or nitrogen source.

The metabolic demand for a given sample was calculated as follows. First, we calculated an essentiality score for each metabolite with transport reactions for each of the 25 genera. The essentiality score for a given metabolite was defined as the fraction of *in silico* media conditions in which the metabolite was essential for biomass synthesis. To simulate media conditions for a given reconstruction, we closed all uptake reactions and then randomly opened a random subset of uptake reactions by sampling from a binomial distribution with p = 0.9. If a random media composition thus defined supported biomass synthesis, we tested the essentiality of metabolites with open uptakes in said media. For this, the uptake reaction of the metabolite was closed, and the feasibility of biomass synthesis was evaluated with FBA. If the model was unable to simulate biomass synthesis after closing the uptake of the metabolite, the metabolite was deemed essential for that condition. Altogether, a total of 1000 viable random media were considered for each genus, and the essentiality score of each metabolite was calculated relative to the number of media in which the metabolite was present. Second, the essentiality scores for each genus were averaged across metabolites and multiplied by the core abundance of the genus. Third, we summed the product of core relative abundances and average essentiality scores across core genera. For individual metabolites, the metabolic demand was calculated as the sum across core genera of the essentiality scores for the metabolite multiplied by core relative abundances.

The 90% probability for external metabolites used to simulate random media was used to account for differences in metabolite essentiality across environmental conditions. Similar results were obtained at lower metabolite probabilities and when, in addition to metabolites with transporters, reactions for the direct uptake of metabolites without transporters were also included with various probabilities (Supplementary Fig. 8). This was done to account for possible transport reactions missing in the metagenomics annotation.

**Untargeted metabolomics**. For both the control and narasin treatments, 15 cecal samples and 10 serum samples from day 35 birds were sent to Metabolon (Morrisville, NC) for untargeted metabolomics analysis. Serum samples were selected sequentially, that is, without knowledge of bird performance or results from other analyses. Metabolomics data were obtained via Ultrahigh Performance Liquid Chromatography-Tandem Mass Spectroscopy (UPLC-MS/MS) (Supplementary Methods).

Peaks were quantified as area-under-the-curve (AUC) detector ion counts (Supplementary Data 9). For each identified metabolite, AUC values were rescaled by dividing by the median AUC value across all samples. Any missing values after re-scaling were imputed with the minimum value observed for that compound across samples. Correlations between metabolite levels and corresponding bird weights at day 35 were calculated using the Pearson correlation of the log-transformed data.

A classification of metabolites into pathways and super-pathways (Supplementary Data 10) was used for running GSEA based on metabolites ranked by their Welch's t score between treatments and by their correlation with weight. A minimum set size of 3 metabolites per set was used.

**Statistics and reproducibility**. Weekly values for ADG and FCR were compared between AGP treatments and control using the one-sided Mann Whitney U test; comparisons between AGP treatments were done using the two-sided test. In both cases 5 replicates (pens) per treatment group were used. α-diversity measures and metabolic demands between pairs of AGP treatments were compared using the two-sided Mann-Whitney U test. Fifteen replicates (individual birds) per treatment were used for comparison. AMR RPKM values, number of AMR genes and carbon and nitrogen compounds with higher or lower utilization potential were compared using the one-sided Mann Whitney U test (n = 15). Metabolite abundances between treatments in either the cecum or serum were compared using Welch's t tests on the log-transformed scaled and imputed data. Fifteen replicates (individual birds) per treatment were used for cecum samples, whereas 10 replicates were used for serum samples.

**Reporting summary**. Further information on research design is available in the Nature Research Reporting Summary linked to this article.

## Data availability

Raw 16 S and shotgun metagenomics sequences were deposited in the SRA database under accession number PRJNA751585. Draft metabolic reconstructions for the 25 core genera, and taxonomic and functional profiles based on the 16 S and shotgun metagenomics data are available from https://github.com/platyias/BroilerCoreModels.

## Code availability

Python code for flux simulations of nutrient utilization potential and metabolite essentiality are also available from https://github.com/platyias/BroilerCoreModels[81].

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

## Acknowledgements

We thank Jeffrey Escobar Monestel for illuminating discussions and comments on the manuscript.

## Author contributions

G.P., N.T.B., D.S., D.G., A.V., and A.K.M. conceived and designed the study. N.T.B., D.S., and A.V. carried out the clinical study and acquired the data. G.P., A.N., and J.B. performed bioinformatics analyses. G.P. carried out formal analysis. G.P., D.S., N.T.B., A.V., D.G., S.P.M., T.B.H., and A.K.M. analyzed and interpreted the data. T.B.H. and A.K.M. supervised the study. G.P. drafted the manuscript. All authors reviewed and edited the manuscript.

## Competing interests

The authors declare the following competing interests: This work was funded by Elanco. All authors were Elanco employees when the study was completed. Elanco sells products containing some of the antibiotics used as treatments in this study.
