## [Peer Review File · Communications Biology]

Reviewers' comments:

Reviewer #1 (Remarks to the Author):

This manuscript provides interesting information related to the impact of the use of AGPs on the intestinal microbiome composition and functions. Furthermore, the manuscript describes the different metabolic and taxonomic changes caused by 4 AGPs. There is a special focus on the use of narasin and its impact on the use of different nitrogen sources by the microbial population in the ceca. This is a topic of high relevance at the moment due to the presentation of AMR and the decrease in the use of antibiotics in animal production. Therefore, it is necessary to understand how possible alternatives should work to be economically viable in relation to animal production parameters.

The level of detail in the manuscript supports the hypotheses and conclusions provided by the authors. I'm providing additional comments below to help to improve the quality of the manuscript or obtain further clarification of possible ideas or assumptions.

Additional comments:

Line number

Line 19 – Narasin is the most effective AGP to improve performance? Or what is the parameter that you are referring to?

Line 19 – Did you check if the AGPs have an effect on the inflammatory/immunological response of the animals? That is one of the principal hypotheses related to the MOA of AGPs.

Line 27 – Are you referring to antimicrobial-resistant infections in humans? Or what do you mean by clinical settings? Or animal clinical studies?

Line 29-30 – Are you referring to a specific geographical location? In the case of Europe, AGPs are completely banned and production without AGPs is increasing worldwide. Therefore, I do not think this can be an affirmation. (Especially when more than 50% of poultry in the US is produced as NAE according to Agristats 2019).

Line 77 – Did you have more than one control group? Or maybe is a typo "controls".

Line 78 – Do you mean "Figures 1A, B"?

Line 78 – As a suggestion: Include the information related to performance parameters (ADG, FI, and FCR) in a table. Unfortunately, it is hard to read the results in Figures 1A and 1B.

Line 78 – Based on Figure 1A and 1B, no statistical differences were observed between AGP?

Line 78 – Based on Figure 1A and 1B, a tendency to increase the value of FCR was observed from day 14 to day 28. Nevertheless, this trend is not observed at day 35 (values appear to be similar to day 28). Do you have an explanation for this behavior of the data? Usually, the FCR increases with age (as observed from days 14, 21, and 28).

Line 80 – Are the FCR values in figure 1B: 0 to 7, 0 to 14, 0 to 21, 0 to 28 and 0 to 35? Just checking if this is the correct interpretation.

Line 82 – Was the difference observed with Narasin significantly different from the other AGP?

Line 85 – The lines presented in each bar in figure 1A, 1B, 1E, and 1F represent the standard error or the standard deviation?

Line 163 – "controls"? Typo?

Line 165 – 171 – This information is quite confusing.

Line 177 – Do you have a hypothesis related to the positive correlation between urea degradation and body weight in broilers?

Line 199 – The color of P-value 0.10 and P-value 0.01 is almost the same, making it difficult to differentiate these values on the graph.

Line 244 – "controls"? Typo?

Line 311 – The control group was used as a baseline? Is that the reason it is not present on the graph (5C, D)? Just to clarify.

Line 317-318 – Check punctuation.

Line 339 – Did you select narasin based on performance benefits? Or do you have any other considerations?

Line 357 – To be consistent with the graph, change urate by uric acid

Line 378 – Replace "organs" to avoid referring to blood as an organ.

Line 396-399 – How stable is the gut microbiome in commercial conditions? I think this is actually

not a really accurate assumption. There are multiple factors that affect this and production farms are actually not equal (feed, environment, litter, management). Please support this assumption or provide an explanation.

Line 403 – What was the criteria to select these four AGP?

Line 415 – Have you compared these results with work from other researchers?

Line 500 – Based on the behavior of chickens – It is probable that the results of the study are influenced by the microbial composition of the litter. Especially when birds are raised on used litter. – Did you check the microbiome of the litter?

Line 504 – Why did you use this kind of test? Non-normal distribution?

Line 505 – Why did you select these sampling dates? Just curious

Line 668 – Each organ refers to “Each ceca sample”?

Line 959 – There is no description of “supplementary Fig. 8A or 8B” in the text

It will be really interesting to see if a replicate of this study under almost identical conditions shows similar results because the lack of consistency is one of the major obstacles in studies related to the microbiome. Maybe increasing the importance of metabolomic evaluations will provide a higher insight into the microbial impact on host performance.

Reviewer #2 (Remarks to the Author):

This manuscript performed a systematic investigation of antimicrobial growth promoters (AGP) and studied the relationship among AGP, host growth and metabolic phenotypes, community and functional variation of microbiota, and antibiotic resistance genes. The authors charted the landscape of microbiome functional changes induced by antibiotics as well as other types of nutritional interventions. These themes are relatively novel in the microbiome field and their results and conclusions are reliable. The manuscript was well written and described. However, there are some questions requiring answers.

Main question:

1. The authors should give detailed info on antimicrobial growth promoters (AGP), e.g., manufacturers, dosage form, dose, and standard dose (or recommended dose) of each AGP, and make a discussion of why this dose standard is reasonable.
2. The authors should give species and gene-set profiles of the microorganism annotated from 16S and metagenomes data.
3. Line 215. As far as I know, correlation coefficients whose magnitude are less than 0.5 indicate variables that have a low correlation. The author needs more explanation to prove the credibility of the conclusion here.
4. Line 237. If they used AMR++ 2 or MEGARes 2.0, maybe they should cite this reference: Doster, E., Lakin, S. M., Dean, C. J., Wolfe, C., Young, J. G., Boucher, C., Belk K. E., Noyes N. R., Morley P. S. (2019) MEGARes 2.0: a database for classification of antimicrobial drug, biocide and metal resistance determinants in metagenomic sequence data. *Nucleic Acids Res.* doi:10.1093/nar/gkz1010.

Minor question:

1. The authors should use a uniform description of “P-value” (in this manuscript, there are three forms, e.g., “p”, “p-value”, “P-value”. Are they represent the same meaning of “P-value”?)
2. Line 157, antimicrobial resistance (AMR) had been annotated before (line 26).

Reviewer #3 (Remarks to the Author):

Manuscript contains interesting findings and is well written. The following comments would help increase the readability of the manuscript.

Five replicates per treatment for growth and gut microbiota indices seem to be low to see their association. It might be better if the treatments had more than 7 replications. Please provide the validity of replications used in this study with statistical power.

The experiment used the used litter. Used litter is very complicate. It needs to have more explanation. How sure that the used litter had equal quality with respect to microbiota?

L491: Specify how chicks were sexed.

Figure 1B. Feed conversion ratio at 7 days was higher. In general, the FCR at first week is the lowest. Something happened at 7 days?

L426: Higher arginase would be due to higher arginine absorption systemically. Thus, higher urea in the control group might not be higher arginase activity. Maybe, it relates to metabolic consequences or amino acid oxidation which might increase nitrogen excretion.

L445-465: Could these findings link to alteration in microbiome or microbiota at species or genus levels? In general, Discussion lacks the role of individual microbiota at species, genus, or strain levels. It is not clear how dietary AGP affect them at the individual level. Many studies with dietary AGP have been published with their impact on gut microbiota in chickens, but the current study lacks those references in Discussion.

Reviewers' comments:

Reviewer #1 (Remarks to the Author):

This manuscript provides interesting information related to the impact of the use of AGPs on the intestinal microbiome composition and functions. Furthermore, the manuscript describes the different metabolic and taxonomic changes caused by 4 AGPs. There is a special focus on the use of narasin and its impact on the use of different nitrogen sources by the microbial population in the ceca. This is a topic of high relevance at the moment due to the presentation of AMR and the decrease in the use of antibiotics in animal production. Therefore, it is necessary to understand how possible alternatives should work to be economically viable in relation to animal production parameters.

The level of detail in the manuscript supports the hypotheses and conclusions provided by the authors. I'm providing additional comments below to help to improve the quality of the manuscript or obtain further clarification of possible ideas or assumptions.

We thank the reviewer for the thoughtful comments, questions, and suggestions. These have helped us improve the clarity of the manuscript.

Additional comments:

1. Line 19 – Narasin is the most effective AGP to improve performance? Or what is the parameter that you are referring to?

The reviewer is correct, we rank Narasin as the most effective AGP based on the mean values for average daily gain and feed conversion ratio compared to the remaining treatments. We have modified the abstract to reflect that we are referring to the effect on increasing performance (line 16).

2. Line 19 – Did you check if the AGPs have an effect on the inflammatory/immunological response of the animals? That is one of the principal hypotheses related to the MOA of AGP's.

Unfortunately, we did not measure any specific immune markers that would indicate a direct effect of AGPs on inflammation. We do however discuss changes in the abundance of metabolites with anti-inflammatory properties such as bilirubin, as well as differences in the prevalence of matrix polysaccharide degradation genes in the cecal microbiome, both of which are consistent with a reduced inflammatory response in narasin treated birds compared to control birds (lines 412-421).

3. Line 27 – Are you referring to antimicrobial-resistant infections in humans? Or what do you mean by clinical settings? Or animal clinical studies?

We have modified this sentence to reflect concerns about AMR selection leading to antimicrobial resistant infections in both humans and animals (line 25).

4. Line 29-30 – Are you referring to a specific geographical location? In the case of Europe, AGP's are completely banned and production without AGPs is increasing worldwide. Therefore, I do not think this

can be an affirmation. (Especially when more than 50% of poultry in the US is produced as NAE according to Agristats 2019).

The reviewer is correct, while NAE production is increasing and AGPs have been banned in Europe, the rapid adoption of intensive farming in countries such as Brazil, Russia, India, China, and South Africa has led to projections of overall increased antimicrobial use by 2030 (See Van Boeckel et al. PNAS. 112:5649, 2015). We have edited this sentence to clarify the effect of trends in specific geographies (line 28).

5. Line 77 – Did you have more than one control group? Or maybe is a typo “controls”.

There was a single control group, we have now corrected this throughout the text.

6. Line 78 – Do you mean “Figures 1A, B”?

Yes, we have indicated this in the text.

7. Line 78 – As a suggestion: Include the information related to performance parameters (ADG, FI, and FCR) in a table. Unfortunately, it is hard to read the results in Figures 1A and 1B.

We thank the reviewer for this suggestion, we now present the corresponding data as a table in Supplementary Data 1. The raw data underlying both figures is also available as source data (Supplementary Data 24)

8. Line 78 – Based on Figure 1A and 1B, no statistical differences were observed between AGP?

We have now included results for the comparison between AGP in Supplementary Data 1. There are only a few comparisons with a p-value < 0.05. These correspond to a lower FCR of narasin compared to BMD on days 14 to 21 and 28 to 35. We have updated Figure 1B to reflect these results.

9. Line 78 – Based on Figure 1A and 1B, a tendency to increase the value of FCR was observed from day 14 to day 28. Nevertheless, this trend is not observed at day 35 (values appear to be similar to day 28). Do you have an explanation for this behavior of the data? Usually, the FCR increases with age (as observed from days 14, 21, and 28).

As described by the reviewer we observe an increase in FCR between day 14 and day 28 regardless of treatment. While FCR is expected to continue increasing during the finisher phase, this wasn't the case. Unfortunately, we do not have a good explanation for why FCR leveled-off in this study, nor do we have other studies looking at the same time periods under similar conditions for comparison. It is possible that an increase in FCR would have been observed had the study continued to day 42.

10. Line 80 – Are the FCR values in figure 1B: 0 to 7, 0 to 14, 0 to 21, 0 to 28 and 0 to 35? Just checking if this is the correct interpretation.

We apologize for the confusion, the FCR values are calculated for each week of the study (0 to 7 days, 7 to 14, 14 to 21, 21 to 28 and 28 to 35). We have modified the x-axis in figures 1a and 1b to make this clear.

11. Line 82 – Was the difference observed with Narasin significantly different from the other AGP?

As we now indicate in Figures 1b and Supplementary Data 1, narasin showed significantly improved performance compared to BMD during at least one of the weeks of the study. The average performance with narasin was in general numerically higher than that of the remaining AGP. Given the consistency of this observation both for AGP and FCR throughout the study, it is likely that the lack of significance between AGP is due to the limited number of replicates used to measure performance (5 pens per treatment).

12. Line 85 – The lines presented in each bar in figure 1A, 1B, 1E, and 1F represent the standard error or the standard deviation?

The lines in the boxplots in figures 1a, 1b, 1e and 1f represent the range of the data, excluding outliers. Outliers were defined as datapoints at least 1.5 times the interquartile range above Q3 or below Q1. We have included a description of the boxplots in the legend of figure 1.

13. Line 163 – “controls”? Typo?

Fixed.

14. Line 165 – 171 – This information is quite confusing.

We have modified the text to facilitate reading of the results (lines 164-173). Briefly, we compared the results of GSEA after ranking gene functions either by 1) their difference in abundance between each of the AGP treatments and the control, or 2) the correlation of their abundance with bird weight at day 35 within each treatment. We show that the gene functions enriched by both methods overlap in the expected direction and provide several examples.

15. Line 177 – Do you have a hypothesis related to the positive correlation between urea degradation and body weight in broilers?

The observation of higher levels of urea degradation genes in birds with higher body weights is one interesting finding of our study. As referenced in the discussion (lines 395-398), experiments have shown that labeled urea injected directly in the cecum is re-absorbed as protein, amino acids, and urea. Thus, it is possible that nitrogen re-cycling of otherwise excreted urea contributes to weight gain in birds with higher urea degradation activity. In addition, the differences in the levels of urea degradation genes with narasin were accompanied by a lower abundance of urea in serum compared to control birds. Urea is potentially toxic, thus increased degradation in the cecum, or lower production by the host, could translate to reduced re-absorption and lower toxicity.

16. Line 199 – The color of P-value 0.10 and P-value 0.01 is almost the same, making it difficult to differentiate these values on the graph.

We thank the reviewer for pointing this out. We have modified the color scheme so that the differences are clear.

17. Line 244 – “controls”? Typo?

Fixed.

18. Line 311 – The control group was used as a baseline? Is that the reason it is not present on the graph (5C, D)? Just to clarify.

The reviewer is correct, the panels indicate the number of carbon and nitrogen compounds with a significantly higher or lower predicted utilization potential compared to the control group.

19. Line 317-318 – Check punctuation.

Checked.

20. Line 339 – Did you select narasin based on performance benefits? Or do you have any other considerations?

In addition to being the AGP with the highest average impact on performance, narasin resulted in the largest difference in both microbiome composition and gene functions compared to the control group. Thus, we anticipated it would also be the most likely to show differences in the cecal and serum metabolome.

21. Line 357 – To be consistent with the graph, change urate by uric acid

The figure was updated for consistency.

22. Line 378 – Replace “organs” to avoid referring to blood as an organ.

The word organs has been replaced by “body sites” to address the reviewer’s point.

23. Line 396-399 – How stable is the gut microbiome in commercial conditions? I think this is actually not a really accurate assumption. There are multiple factors that affect this and production farms are actually not equal (feed, environment, litter, management). Please support this assumption or provide an explanation.

We appreciate the reviewer’s comment and agree that the composition of the cecal microbiome changes in response to factors that vary both within and across production farms. We note that the cecum microbiome is especially stable during a single growth cycle within specific farms. This is illustrated by the fact that even across AGP treatments, age remains the main contributor to the variation in microbiome composition in our study, enabling the detection of differences associated with treatments without being dominated by inter-individual variation. This is also supported by the work by Johnson et al. (Appl Environ Microbiol. 84: e00362-18, 2018) discussing how even across farms in different geographical locations, the flock-to-flock microbiome variation is smaller than the

variation across flock cycles. To avoid confusion and in light of space restrictions, we have removed this reference to cecal microbiome stability from the discussion, but have kept it in the introduction (line 54).

24. Line 403 – What was the criteria to select these four AGP?

The four AGP considered in our study were selected 1) based on their reported use as growth promoters, and 2) to encompass different antimicrobial classes with distinct antibacterial modes of action (inhibition of protein synthesis, inhibition of cell wall synthesis, or increasing membrane permeability). These considerations allowed us to investigate whether at the level of the cecal microbiome the four antimicrobials had similar or distinct effects and potential microbiome-mediated functional impacts on performance.

25. Line 415 – Have you compared these results with work from other researchers?

While we have not come across similar analyses, tying the metabolic requirements for growth of intestinal bacteria to host performance, several observations align with results from other groups. For example, a lower concentration of amino acids in the cecum was also recently observed by Johnson et al. (Scientific Reports 9:8212, 2019) in turkeys fed with BMD. Our observations regarding lipid concentrations in the cecum together with the lower abundance of bile salt hydrolases under AGP aligns with previous research suggesting altered lipid metabolism as an AGP mechanism (e.g. Guban et al. Poultry Sci. 85:2186, 2006). Finally, other studies comparing more than one AGP found different effects on the composition of the cecal microbiota, which is consistent with our results. We now include these references in the discussion (lines 427-432)

26. Line 500 – Based on the behavior of chickens – It is probable that the results of the study are influenced by the microbial composition of the litter. Especially when birds are raised on used litter. – Did you check the microbiome of the litter?

Indeed, the composition of the cecal microbiome has been shown to be affected by the quality of litter used. It is thus possible that the differences observed following AGP treatment would not be the same if a different litter was used. We did not check the litter microbiome, but it would be interesting to investigate what is the impact of different microbial exposures on the microbiome modulating effect of AGPs and their impact on performance. We now point to this in a limitations paragraph in the discussion (line 422).

27. Line 504 – Why did you use this kind of test? Non-normal distribution?

We opted for the Mann-Whitney U test in order to not make assumptions about the underlying distribution. In particular, given that the ADG and FCR values compared included only 5 replicates per treatment. Nevertheless, we note that the vast majority of significant comparisons according to the non-parametric test were also significant using a t-test.

28. Line 505 – Why did you select these sampling dates? Just curious

The sampling dates of 7, 21 and 35 days of age were selected based on the diet schedule. Specifically, we wanted to look at differences in the gut microbiome through bird development; given that different diets were fed between days 0-14, 14-28 and 28-35, it seemed appropriate to collect samples while animals were fed each of the three diets.

29. Line 668 – Each organ refers to “Each ceca sample”?

The phrase refers to each body site (cecum or serum), we have modified the text to clarify this.

30. Line 959 – There is no description of “supplementary Fig. 8A or 8B” in the text

Supplementary figures 8a and 8b are referenced in the methods section describing the prediction of essential metabolites for core bacterial genera (line 624). We added an additional reference (line 296) in the results to highlight the robustness of the method to the selected parameters.

31. It will be really interesting to see if a replicate of this study under almost identical conditions shows similar results because the lack of consistency is one of the major obstacles in studies related to the microbiome. Maybe increasing the importance of metabolomic evaluations will provide a higher insight into the microbial impact on host performance.

We agree that replicability of observed differences across microbiome studies is one important challenge. By studying not only the taxonomic composition, but also including gene function and metabolomics profiles in the analysis, our study provides additional dimensions for reproducing and validating possible mechanisms through which the microbiome impacts performance. We now discuss this in the manuscript (lines 425-427).

Reviewer #2 (Remarks to the Author):

This manuscript performed a systematic investigation of antimicrobial growth promoters (AGP) and studied the relationship among AGP, host growth and metabolic phenotypes, community and functional variation of microbiota, and antibiotic resistance genes. The authors charted the landscape of microbiome functional changes induced by antibiotics as well as other types of nutritional interventions. These themes are relatively novel in the microbiome field and their results and conclusions are reliable. The manuscript was well written and described. However, there are some questions requiring answers.

We thank the reviewer for the comments and suggestions.

Main question:

1. The authors should give detailed info on antimicrobial growth promoters (AGP), e.g., manufacturers, dosage form, dose, and standard dose (or recommended dose) of each AGP, and make a discussion of why this dose standard is reasonable.

We have now included Table 1 in the Methods section with the information suggested by the reviewer. Briefly, the doses were picked based on informal polling of several veterinarians and nutritionists asked about commonly used doses for growth promotion. Importantly, all of the doses used fall within dose ranges indicated either now, or previously, in product labels. This is now described in the manuscript (lines 454-456).

2. The authors should give species and gene-set profiles of the microorganism annotated from 16S and metagenomes data.

We thank the reviewer for the suggestion. The ASV counts and taxonomic assignments from the 16S data, as well as species, genus and gene functional role profiles from the shotgun metagenomics data are now available from <https://github.com/platyias/BroilerCoreModels>, together with the corresponding metadata of both datasets.

3. Line 215. As far as I know, correlation coefficients whose magnitude are less than 0.5 indicate variables that have a low correlation. The author needs more explanation to prove the credibility of the conclusion here.

The reviewer is correct that most correlations observed are low. Nevertheless, this is expected given that the relative abundance of a specific genus in a sample is calculated based on tens of thousands of reads classified to reference genomes, while the abundance of genes with a specific function is often based on just tens or at best hundreds of reads. The lower read coverage at the gene level should introduce some stochasticity in the correlations. The fact that for most core genera the correlations are positive and significant shows that despite some genera having very low abundances, and their gene abundance estimates being noisier, their relative abundance and contribution to enriched gene functions are not independent. Moreover, in addition to the correlations, we also evaluated the overlap between genes annotated to core genera based on samples from different treatments. The high overlap observed (~80%), suggest that the contribution of core genera to functional differences between treatments are likely due to abundance differences. We have included an additional note regarding the low correlations observed (lines 215-218).

4. Line 237. If they used AMR++ 2 or MEGARes 2.0, maybe they should cite this reference: Doster, E., Lakin, S. M., Dean, C. J., Wolfe, C., Young, J. G., Boucher, C., Belk K. E., Noyes N. R., Morley P. S. (2019) MEGARes 2.0: a database for classification of antimicrobial drug, biocide and metal resistance determinants in metagenomic sequence data. Nucleic Acids Res. doi:10.1093/nar/gkz1010.

We have updated the corresponding reference in the manuscript.

Minor question:

5. The authors should use a uniform description of "P-value" (in this manuscript, there are three forms, e.g., "p", "p-value", "P-value". Are they represent the same meaning of "P-value"?)

We thank the reviewer for pointing this, we have corrected the text to use a consistent wording for p-values.

6. Line 157, antimicrobial resistance (AMR) had been annotated before (line 26).

We thank the reviewer for catching this.

Reviewer #3 (Remarks to the Author):

Manuscript contains interesting findings and is well written. The following comments would help increase the readability of the manuscript.

We thank the reviewer for the comments, which have contributed to improving the manuscript.

1. Five replicates per treatment for growth and gut microbiota indices seem to be low to see their association. It might be better if the treatments had more than 7 replications. Please provide the validity of replications used in this study with statistical power.

The reviewer is correct in pointing out that comparisons at the level of pens (5 per treatment) could have led to missed differences due to low statistical power. The reason we focused on pens and not individual birds was to show results for both average daily gains and feed conversion ratios (feed weigh-backs for calculating FCR were done weekly at the pen level). Based on the observed values for ADG during the final week of the study, we retrospectively estimate the statistical power of a one-sided Mann-Whitney U test with n=5 to be between 0.71 and 0.98 for the AGP treatments most similar and most different from the control, respectively. Similarly, we estimate power to be between 0.5 and 0.83 for FCR. These results are based on the procedure by Mollan et al. (2019, arXiv:1901.04597) implemented in the wmwpowd method of the wmwpow R package (v. 0.1.3).

For individual birds, weights were recorded for, on average, 68 birds per treatment on day 35. Weight comparisons of individual antibiotic treated birds to control birds were consistent with results at the pen level, with p-values ranging between 10^{-3} and 10^{-8} .

Importantly, for the comparisons of microbiome taxonomic and functional profiles, analyses were performed at the level of individual birds, with 15 birds sampled per treatment. Significant differences in community structure between treatments (based on ANOSIM at the species level) similar to those reported in the manuscript were still detectable when considering as few as 7 randomly selected samples per treatment.

2. The experiment used the used litter. Used litter is very complicate. It needs to have more explanation. How sure that the used litter had equal quality with respect to microbiota?

We used litter from untreated (control) birds from a previous study in an adjacent barn at the same facility. The litter was mixed 50/50 with fresh litter and distributed to the pens in this study. To prevent any potential differences in qualities of the litter from impacting the conclusions of the study, pens were randomly assigned to treatments such that any differences in litter between pens were independent of treatment group. In addition, upon bootstrapping the 16S data of pens assigned to each treatment (100 bootstrap runs), we verified similar differences in microbial community composition between treatments for the day 35 animals in >80% of cases. Indicating that observed differences were not driven by specific pens with unique properties, such as a particular litter quality. We have now extended the description of the litter used in the study (Line 457).

3. L491: Specify how chicks were sexed.

We have added a description in the Methods (line 448). Chicks were sexed by the hatchery by feather sexing.

4. Figure 1B. Feed conversion ratio at 7 days was higher. In general, the FCR at first week is the lowest. Something happened at 7 days?

The reviewer is correct. We now address this in the text (line 81). The explanation for the unexpectedly high FCR during the first week is that during this period feed was offered to the birds in trays on the floor. This was done to facilitate access to feed, but it also makes it easy for food to be kicked or carried out of the tray leading to artificially higher feed intake estimates.

5. L426: Higher arginase would be due to higher arginine absorption systemically. Thus, higher urea in the control group might not be higher arginase activity. Maybe, it relates to metabolic consequences or amino acid oxidation which might increase nitrogen excretion.

This is an interesting point. The review by Ball et al. (J Nutr 137:1626S, 2007) cited in the manuscript explicitly describes how excreted urea in chicks is a measure of arginase activity, which occurs primarily in the kidneys. Thus, while the reviewer is correct pointing out that higher arginase activity would correlate with increased arginine absorption, it would also simultaneously increase the levels of excreted urea, which is a direct byproduct of arginase. Both effects were coupled and enhanced in experiments carried under arginine-limited diets in the cited study by Austic and Nesheim (J Nutr 100:855, 1970). Urea is not considered a major nitrogen excretion product in birds due to the low expression of urea cycle enzymes in the liver. Nevertheless, differences in amino-acid metabolism could indeed contribute to the observed result by limiting the availability of essential amino acids such as arginine.

6. L445-465: Could these findings link to alteration in microbiome or microbiota at species or genus levels? In general, Discussion lacks the role of individual microbiota at species, genus, or strain levels. It is not clear how dietary AGP affect them at the individual level. Many studies with dietary AGP have been published with their impact on gut microbiota in chickens, but the current study lacks those

references in Discussion.

We appreciate this comment from the reviewer. We show in the manuscript that changes in the abundance of specific gene functions upon AGP treatment can often be explained by genes mapped to a handful of genera that were prevalent across samples. It follows that abundance (or gene content) differences at the genus or species level would underlie the observed functional differences. For example, we describe how the potential for urea utilization in the cecum under narasin treatment is likely associated with higher abundances of Ruminococcus, Blautia, and Lachnospirillum; we also show that differences in AMR and virulence genes with virginiamycin are associated with the abundances of Salmonella, Escherichia, and Staphylococcus.

As the reviewer correctly points out, several studies have investigated taxonomic differences (primarily at the genus level) upon treatment with one or a few AGP (e.g. references 33, 35, 36, 37). To complement our study, we now present Supplementary Data 2 showing the results of a genus-level differential abundance analysis for the day 35 samples (Line 136). The results reflect trends observed in our other analyses, with more differentially abundant genera in the virginiamycin and narasin treatments relative to control, compared to BMD and avilamycin.

REVIEWERS' COMMENTS:

Reviewer #1 (Remarks to the Author):

Appreciate the time provided by the authors to reply to all my comments and also make the corresponding manuscript modifications. The manuscript is well-written and presents interesting information related to the impact of the inclusion of 4 different AGPs in poultry diets. The conclusions are well supported based on the data generated and analyzed during the study. A couple of suggestions were also provided for future studies including measurement of litter microbiome and immune profile of the animals under study.

Reviewer #3 (Remarks to the Author):

Authors addressed the comments raised by the reviewer.